# Cardiometabolic Disorders Are Important Correlates of Vulnerability in Hospitalized Older Adults

**DOI:** 10.3390/nu15173716

**Published:** 2023-08-25

**Authors:** Ganna Kravchenko, Renata Korycka-Bloch, Serena S. Stephenson, Tomasz Kostka, Bartłomiej K. Sołtysik

**Affiliations:** Department of Geriatrics, Healthy Aging Research Centre (HARC), Medical University of Lodz, Haller Sqr. No. 1, 90-647 Lodz, Poland; ganna.kravchenko@umed.lodz.pl (G.K.); renata.korycka-bloch@umed.lodz.pl (R.K.-B.); serena.stephenson@umed.lodz.pl (S.S.S.); bartlomiej.soltysik@umed.lodz.pl (B.K.S.)

**Keywords:** elderly, VES-13, frailty, multimorbidity, cardiovascular diseases, diabetes

## Abstract

With an increasingly aging population worldwide, the concept of multimorbidity has attracted growing interest over recent years, especially in terms of frailty, which leads to progressive multisystem decline and increased adverse clinical outcomes. The relative contribution of multiple disorders to overall frailty index in older populations has not been established so far. This study aimed to assess the association between the vulnerable elders survey-13 (VES-13) score, which is acknowledged to be one of the most widely used measures of frailty, and the most common accompanying diseases amongst hospitalized adults aged 60 years old and more. A total of 2860 participants with an average age of 83 years were included in this study. Multiple logistic regression with adjustment for age and nutritional status was used to assess the independent impact of every particular disease on vulnerability. Diabetes mellitus type 2, coronary artery disease, atrial fibrillation, heart failure, chronic kidney disease, osteoarthritis, fractures, eyes disorders, depression, dementia, pressure ulcers, and urinary incontinence were associated with higher scores of VES-13. Hospital admission of older subjects with those conditions should primarily draw attention to the risk of functional decline, especially while qualifying older patients for further treatment in surgery and oncology. At the same time, lipid disorders, gastrointestinal diseases, higher body mass index, and albumins level were related to a lower risk of being vulnerable, which may be attributed to a younger age and better nutritional status of those patients.

## 1. Introduction

The global population is aging, and this demographic shift poses a significant challenge for healthcare systems worldwide. Older adults are more likely to develop multiple chronic conditions, which can increase their risk of functional decline and death.

To address this challenge, healthcare professionals need accurate tools to predict the risk of functional decline and death in older patients. One of the tools used is the vulnerable elders survey-13 (VES-13) which is a brief, self-administered questionnaire. It assesses various domains of health, including activities of daily living, physical function, and self-rated health, and it is one of the most widely-used measures of frailty [1,2].

Research shows that VES-13 is a valuable tool for identifying older adults with a limited life expectancy [3]. Nevertheless, although VES-13 was developed for a 2-year estimation of functional decline and mortality risks, it might be useful in a 5-year prediction period as well [4]. High scores of VES-13 are associated with lower quality of life. Silva et al. reported that VES-13 shows 84% sensitivity and 98.2% negative predictive value while testing the quality of life in older individuals. Additionally, according to this study, VES-13 might help to determine which patients require comprehensive geriatric assessment [5].

There are many publications on the usage of VES-13 in oncology. The scale of VES-13 showed a prominent level of reliability and operability in patients with advanced castration-resistant prostate cancer [6], gastrointestinal cancer [7], and gynecological malignancies [8]. Recent research reported that vulnerability predicts mortality better than a diagnosis of prostate cancer among older men [9]. Besides that, VES-13 might be helpful to assess older oncological patients who are at risk of chemotherapy toxicity and help with choosing a method of treatment [8,10].

An interesting study showed that VES-13 might be useful in the preoperative identification of frailty and vulnerability and in assisting perioperative decision-making in geriatric patients. VES-13 scores were correlated to hospital length of stay, post-anesthesia care unit stay, altered mental status, and morbidity [11]. In older adults with traumatic injuries, each additional VES-13 point was associated with a greater risk of complication or death [12]. In patients 60 years and older, who were hospitalized with COVID-19 infection, VES-13 was performed to estimate the risks of in-hospital mortality and the need for invasive mechanical ventilation. According to research, patients classified as extremely vulnerable (8–10 points in VES-13 score) were strongly and independently associated with a higher relative risk of those outcomes [13].

Cardiometabolic diseases remain leading causes of death and disabilities, especially in older adults [14]. Coronary artery disease (CAD) and stroke hold the first two positions in the structure of mortality, the second and the third place as causes of disability. Diabetes mellitus type 2 (DM2) is one of the top ten causes of both mortality and disability structure. A prospective observational cohort study was performed in patients hospitalized with acute coronary syndrome and/or decompensated heart failure (HF). A high prevalence of vulnerable patients was found (54%) and vulnerability was associated with increased use of healthcare services, including more clinic visits, emergency room visits, and hospitalizations [15]. Another research found that 44.4% of the patients aged 65 years old or more that were hospitalized for acute coronary syndromes and/or acute decompensated HF were vulnerable at baseline and 34.4% were vulnerable at 90 days after hospital discharge [16].

While numerous data were found in the fields of oncology and surgery, there is a knowledge gap regarding the cardiometabolic diseases’ impact on VES-13. There are studies showing the relationship between single conditions and vulnerability/frailty measured with different tools. Nevertheless, the literature lacks studies on the complex association and contribution of different diseases to overall vulnerability in a “real word” large population of geriatric patients. Identifying the diseases that are most strongly connected with vulnerability is essential for healthcare professionals as they could develop targeted interventions to improve patients’ outcomes.

Therefore, in the present study, we tried to identify which chronic disorders of a geriatric population are associated with vulnerability assessed by VES-13, focusing on cardiometabolic diseases.

## 2. Material and Methods

### 2.1. Patients

For this study, patients were recruited from January 2012 to December 2019 at Central Veterans Hospital in Lodz, Poland. From 2020 to 2022, the department served partially as a COVID-19 ward. In the present study, all 3382 older patients that had been admitted to the Department of Geriatrics from 2012 to 2019 were screened. The criterion for participation in this study was efficient verbal communication. Out of the 3382 patients, 128 were readmitted to the Department of Geriatrics and were excluded from further analyses. A further 394 patients were excluded because of severe dementia or terminal illness. After screening, 2860 patients (2036 women and 824 men) met the inclusion criteria and were enrolled in the analysis.

### 2.2. Anamnesis

Upon admission, detailed medical anamnesis and examination of medical history were performed. Furthermore, a comprehensive geriatric assessment was conducted with each patient. The presence of arterial hypertension (HT), DM2, lipid disorders, current or previous stroke, CAD, current or previous myocardial infarction (MI), atrial fibrillation (AF), chronic HF, chronic kidney disease (CKD), obstructive lung diseases (chronic obstructive pulmonary disease (COPD) and asthma), osteoarthritis (OA), osteoporosis, current or previous fractures, gastrointestinal diseases (chronic gastritis, gastrointestinal ulcer), neoplastic diseases, eyes disorders (glaucoma, cataract), depression, dementia, pressure ulcers (PU), and urinary incontinence (UI) was scrutinized.

### 2.3. Measurements

A VES-13 scale in the Polish version [17] was completed with all patients. The score ranged from 0 to 10, with higher scores indicating a higher risk of functional decline and death. According to research [1], a score of more than three points was associated with a four times higher risk of death or functional decline when compared to elders scoring three or less. Taking that into consideration, a cutoff point of three points was suggested by the authors and used in our study.

Body mass index (BMI) was calculated by dividing the weight (in kilograms) by height squared (in meters). CKD was defined as a glomerular filtration rate (GFR) lower than 60 mL/min/1.73 m^2^ according to BIS1 formula [18]. Depression was assessed using the geriatric depression scale [19], where a score of five or higher indicates the presence of depression. Dementia was diagnosed based on the results of the mini-mental state examination, where a score of less than 24 points indicated the presence of dementia [20]. Albumins concentration was measured in g/L using a 5-Diff Sysmex XS-1000i hematological analyzer (Sysmex, Kobe, Japan).

### 2.4. Statistical Analysis

The normality of distribution was verified with the help of the Shapiro–Wilk test. As several variables were not normally distributed, the data have been presented both as mean ± standard deviation and median (interquartile differences from the first (25%) to the third (75%)). The qualitative variables, such as sex and presence of disease, were presented as raw numbers and percentages of the group. Qualitative variables were compared using the Chi-square test and quantitative variables were compared using the Mann–Whitney U-test. Pearson correlation coefficients were employed to calculate the relationship between variables. After dichotomization of VES-13 with a cutoff point set at three points, the multivariate logistic regression was performed. Diseases, which expressed significance in bivariable analysis, were employed in the logistic regression model, including all the conditions significantly related to VES-13 concurrently. To assess the potential influence of diseases on VES-13 simultaneously, two statistical models were constructed. The first one considered sex and BMI, while the second one also included age and albumins level as continuous variables. Statistical significance was set at *p* ≤ 0.05. Statistical analysis was performed using Statistica 13.1.

### 2.5. Ethical Consideration

The study was conducted according to the guidelines of the Declaration of Helsinki and approved by the Ethics Committee of the Medical University of Lodz (approval number: RNN/68/23/KE, 18 April 2023).

## 3. Results

The median age was 83 (77–87) years for both sexes. Table 1 shows the characteristics of 2860 patients according to sex. There was no significant difference between men and women in age, BMI, the prevalence of HT, stroke, CAD, HF, CKD, gastrointestinal diseases, neoplastic diseases, eyes disorders, dementia, and PU. Men had a higher body mass and a higher prevalence of DM2, MI, AF, and obstructive lung diseases. At the same time, women had a significantly higher level of albumins, more points in VES-13 scale, and a higher prevalence of lipid disorders, OA, osteoporosis, fractures, depression, and UI.

The comparison of VES-13 results between participants with and without particular disease is presented in Table 2. For obstructive lung diseases, OA, and eyes disorders, there were no significant differences in VES-13 scores between patients with and without those conditions. Women with HT, DM2, and MI and men with UI had higher scores of VES-13 as compared to their peers without the disease.

For stroke, CAD, AF, HF, CKD, fractures, depression, dementia, and PU, the presence of disease was associated with significantly higher scores of VES-13 in both sexes. At the same time, patients with lipid disorders or gastrointestinal diseases in both sexes and women with osteoporosis or neoplastic diseases had lower VES-13 results than patients without those conditions.

Linear correlation was performed for age and VES-13 with R = 0.587 (*p* < 0.001), BMI and VES-13 with R = −0.11 (*p* < 0.001), and albumins and VES-13 with inverse correlation, R = −0.37 (*p* < 0.001).

Age of patients, according to the presence of the most common disorders, is presented in Table 3. The patients presenting with diseases such as HT (women), stroke (women), CAD, MI, AF, HF, CKD, osteoporosis (men), fractures, eyes disorders (men), dementia, and PU (women) were significantly older, whereas patients with DM2 (men), lipid disorders, osteoporosis (women), and gastrointestinal diseases (women) were significantly younger in comparison to those without disease.

To assess the potential influence of diseases on VES-13 simultaneously, two statistical logistic regression models were constructed with VES-13 dichotomized as ≥3 points vs. <3 points. The first one considered sex and BMI, while the second one also included age and albumins level. In the first model, DM2, CAD, AF, HF, CKD, OA, fractures, eyes disorders, depression, dementia, PU, and UI were related to higher values of VES-13 (increased odds ratio for vulnerability). Lipid disorders, gastrointestinal diseases, and higher BMI were associated with lower values of VES-13 (Figure 1).

In the second model, DM2, stroke, CAD, HF, CKD, OA, fractures, eyes disorders, depression, dementia, UI, and age were related to higher values of VES-13 (increased odds ratio for vulnerability). The higher albumins level was related to lower values of VES-13 (Figure 2).

## 4. Discussion

To the best of our knowledge, this is the first paper presenting the association between a wide range of the most common chronic disorders and vulnerability assessed by VES-13 in a large, hospitalized geriatric population. Simultaneous assessment of all major concomitant disorders with sex and BMI revealed some diseases were independently related to vulnerability: several cardiometabolic conditions, CKD, OA, fractures, eyes disorders, depression, dementia, PU, and UI. Interestingly, lipid disorders, gastrointestinal diseases and higher BMI were related to lower vulnerability. This may be attributed to the lower age of patients presenting these conditions. In a fully adjusted model, after inclusion of age and albumins representing nutritional state, the statistical impact of several conditions became not evident. Nevertheless, DM2, current or previous stroke, CAD, HF, OA, fractures, eyes disorders, depression, dementia, and UI remained associated with vulnerability status of the patients.

### 4.1. Cardiometabolic Disorders

The majority of cardiometabolic conditions were related to higher vulnerability. DM2, current or previous stroke, CAD, and HF retained their contribution to vulnerability in the fully adjusted model.

Our results are consistent with the data showing that hypertensive patients were more often frail compared to robust patients [21], although the abovementioned study did not use VES-13 but instead used the FRAIL scale. There are also recommendations discussing the perspective of treatment for vulnerable subjects with HT [22,23,24]. While there is not a large body of research examining the relationship between HT, sex, and VES-13, we may find some evidence suggesting that hypertensive women may be at a greater risk for functional decline than men [25]. Likewise, we found that women with HT had higher VES-13 scores than did women without HT. At the same time, there were no significant differences in VES-13 results between men with or without HT.

In the presently available literature, no studies have examined the association between VES-13 scores and history of stroke. However, frailty increases the risk of stroke and mortality in patients with AF [26] as well as promotes increased stroke complications, poorer recovery, and weaker response to treatment [15,27]. In our group, patients with current or previous stroke also had significantly higher VES-13 results in the fully adjusted multivariate model.

According to our data, patients with CAD had significantly higher VES-13 scores in both sexes. When using CAD and MI in multiple regression, only CAD was selected as a factor associated with VES-13. Systematic review reported the bidirectional association between frailty and cardiovascular disease [28]. The study of Purser et al. demonstrates that frailty, measured by the gait speed test, was the strongest predictor of mortality in a population with CAD [29].

AF is strongly related to HT and stroke. Our data show that AF is related to vulnerability in both bivariate association and multivariate model. Frailty deteriorates clinical management of AF and increases the risk of adverse outcomes, such as stroke or bleeding [30,31]. Frail AF patients have an increased risk of death, ischemic stroke, and bleeding, and thus demand a holistic and person-tailored approach to their care [32].

Previous research reported that HF is independently associated with increased vulnerability. Both conditions are inseparably related to sarcopenia, inflammation, and global dysfunction, and independently increase the risk of each other [15,33]. It is consistent with our data that patients with HF presented an almost doubled risk of increased VES-13 scores. In the available literature, there is some information about the sex-heterogeneous pattern of frailty occurrence, especially in coexistence with other disorders like HF [34], stroke, DM2, CKD, dementia, or OA [35]. These sex differences are also apparent in our results.

DM2 is associated with an accelerated aging process that manifests as the premature onset of geriatric syndromes and frailty. Progressive muscle and nerve dysfunction, deterioration of cardiometabolic status, and physical function loss caused by DM2 inevitably led to the development of frailty [36,37,38]. Those findings conform to our results that DM2 is an independent predictor of higher values of VES-13 and in adjusted multivariate models. Pre-frailty and frailty are linked to increased morbidity and mortality, as well as higher healthcare utilization, in individuals with DM2 [39].

Unlike other cardiometabolic conditions, lipid disorders were related to lower levels of VES-13. According to logistic regression, the risk of elevated VES-13 is 42% lower among hyperlipidemic subjects. No research comparing the VES-13 scale with lipid disorders was found in available literature. The relation mentioned above may be the effect of two factors. Our subjects with lipid disorders were significantly younger. On the other hand, frailty is inseparably linked with malnutrition, which is in strong reverse relation with lipid disorders.

Similar interpretation may be put forward for BMI association with VES-13. It is consistent with the study, where normal- or underweight frail older participants have a poorer 3-year survival rate [40]. Jayanama et al. have found that being overweight is a protective factor against mortality in moderately/severely frail people and obesity grade 1 may be protective against mortality for people with at least a mild frailty [41]. The abovementioned papers align with our results that higher BMI may be an independent protective factor for low results of VES-13. On the other hand, some studies did not find a relationship between BMI and frailty [42].

### 4.2. Osteoarthritis, Osteoporosis and Fractures

Subjects with rheumatoid arthritis and OA are more likely to have or develop frailty [43]. Frailty may hinder the self-repair of joint structure [44]; reversely, the presence of OA leads to functional capacity decline [45]. In addition to that, chronic pain caused by OA affects the degree of frailty [46]. The currently available literature does not provide any records referring to OA and VES-13. Our data revealed no statistically significant differences observed in the levels of VES-13 among individuals with OA and those without. However, logistic regression indicates that presence of OA is linked with a 53% and 38% (fully adjusted model) higher risk of elevated results of VES-13.

Our data show no association between VES-13 and osteoporosis in multivariate logistic regression. A prospective cohort study by Sternberg et al. indicated that frailty status as defined by the VES-13 predicts a decrease in bone mineral density (BMD) after 1 year of follow-up [47]. In research with 6 years of follow-up, the difference in calcaneal BMD of 1.4 T-score units between subjects with the highest and the lowest frailty scores corresponded to two to three times higher fracture risk, even after adjusting for age, weight, sex, and race [48].

With the growing number of older subjects in society, the complication associated with fractures will continue to increase [49]. In our study, current or previous fracture was significantly associated with elevated VES-13 (OR 2.15), which is concordant with previous reports. According to Min et al., in hospitalized older patients with traumatic injury, each additional VES-13 point was associated with an increased risk of complications or death [12]. Among traumatic patients, increased frailty scores were independently connected with increased mortality [50], risk of complications, and prolonged hospitalizations [51].

### 4.3. Depression, Dementia, Pressure Ulcers and Urinary Incontinence

All these conditions are interrelated and closely associated with advanced age. Vulnerability is strongly related to the presence of depressive symptoms [52]. Furthermore, some scientists indicate a reciprocal interaction between depression and frailty [53,54]. It is consistent with our findings that patients with depression had significantly worse scores of VES-13, also after full adjustment in the multivariate model.

Dementia is inseparably connected with frailty [55]. Each additional point of the VES-13 refers to a worse outcome in patients with dementia [4]. Our results are fully in line with the abovementioned papers. The elderly with dementia had significantly higher VES-13 scores. In logistic regression, dementia influenced the level of VES-13 the most out of all studied disorders with OR = 4.96. Accordingly, cognitive frailty becomes a newly acknowledged dimension of this geriatric syndrome [56,57].

Age-related skin changes, comorbidities, polypharmacy, reduced mobility, inadequate nutrition and hydration, and continence issues strictly lead to the development of PU [58]. This increased vulnerability is present particularly among the physically limited or bedridden older subjects [59]. In our study, PU were significantly associated with worse VES-13 with OR of 2.51.

UI is twice as common in older people with frailty compared to those without [60]. UI is also a predictor of higher mortality rates, particularly in the geriatric population [61]. Currently available literature does not provide the works investigating the association between VES-13 scores and UI. In our study, a multivariate analysis showed a 79% higher risk of increased VES-13 scores among patients with UI, being concordant with available knowledge in that field.

### 4.4. Other Chronic Conditions

According to the systematic review referring to CKD and frailty [62], the incidence of frailty increases with reduced GFR. It is consistent with our analyses that patients with GFR below 60 mL/min/m^2^ had a 2.69-fold higher risk of an increased score of VES-13. Frailty is highly prevalent and connected with adverse outcomes of CKD mortality and end-stage kidney disease [31].

So far, no study has initiated discussion on obstructive lung diseases and VES-13. Frailty, defined by the Fried criteria, was associated with higher severity of airflow limitation, dyspnea, and more frequent exacerbations in the group of 2.142 participants [63]. Another work has presented that the frailty phenotype is associated with an increased risk of non-COPD hospitalizations [64,65], prolonged hospital stays, worse in-hospital mortality, and decreased activity of daily living at discharge. [66]. However, our research showed no significant difference in VES-13 levels in patients with COPD. This may relate to a lower prevalence of this chronic disorder compared to other diseases.

In older adults with gastrointestinal diseases undergoing abdominal surgery, preoperative frailty was associated with postoperative complication incidence and extended hospital stays [67]. This stands in contradiction with our data. Our patients with gastrointestinal diseases had significantly better results on the VES-13 scale. Gastrointestinal disorders are linked with malnutrition and, indirectly, with frailty. This contradiction may be at least partially explained by the fact that our hospitalized women with this type of disorder were significantly younger.

The presently available literature about VES-13 in the majority refers to neoplastic diseases. According to some researchers, VES-13 is a feasible tool for predicting survival and qualifying for oncological treatment [10,64,65,68,69]. In our results, neoplastic diseases did not impact on VES-13 in multivariate design. This contradiction may be explained by the relatively low prevalence of those disorders in our population and by the fact that we enrolled both past and present oncological problems. Subjects successfully treated for neoplasms often present greater health awareness and better functional outcomes.

Our initial analysis did not reveal any significant differences in VES-13 scores between patients with and without eye disorders. However, a subsequent logistic regression analysis demonstrated a significant association between the presence of eye disorders and higher VES-13 scores, with an odds ratio of 1.61. We did not find any publications connected with VES-13 and eye disorders. However, according to some works, frailty itself is associated with vision impairment [70,71,72].

According to previous research, the scores of two widely used nutritional scales (Nutrition Risk Screening 2002 and the subjective global assessment form) were positively associated with the results of VES-13 [73]. This means that a worse nutritional status was associated with vulnerability. Albumins are a well-known biomarker of malnutrition [74]. Therefore, it is consistent with our results that patients with higher levels of albumins had a lower risk of vulnerability.

Several limitations of this study could be addressed in future research. It is a cross-sectional analysis, and monitoring the vulnerability status after the hospitalization would probably bring new information. This study focused on older inpatients in central Poland. Some patients were excluded because of severe dementia or terminal illness. Multiple medical problems were reported, but every one of the concomitant disorders were not scrutinized. Larger multicenter studies in different populations would help in drawing more general conclusions. The relationship between vulnerability and concomitant diseases may also differ in a community or long-term institutional environment. We only used one short vulnerability screening test (VES-13).; other frailty assessment tools might have performed differently. Nevertheless, the strengths of our research are its large sample size and the inclusion of multiple diseases with their joint impact estimations.

## 5. Conclusions

VES-13 is often used to qualify patients for further comprehensive geriatric assessment. Our results can help to identify which accompanying conditions in older hospitalized adults this assessment will be most necessary for. According to our data, DM2, CAD, AF, HF, CKD, OA, fractures, eyes disorders, depression, dementia, PU, and UI were associated with higher scores of VES-13. At the same time, lipid disorders, gastrointestinal diseases, higher BMI, and albumins levels were related to a lower risk of vulnerability.

Considering the importance of VES-13 for predicting negative clinical outcomes in older patients and qualifying them for further treatment, particularly in surgery and oncology, paying attention to conditions associated with higher VES-13 scores, may be useful for clinicians.

## Figures and Tables

**Figure 1 nutrients-15-03716-f001:**
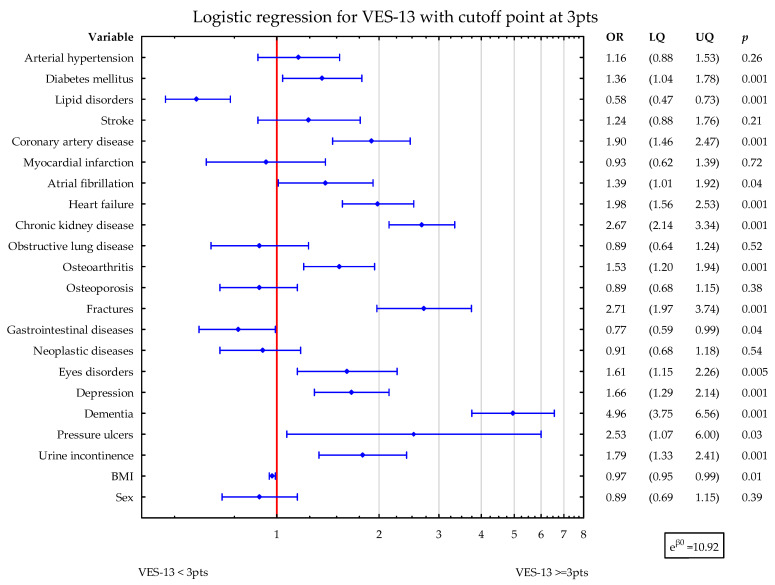
Shows the simultaneous statistical impact of concomitant diseases, BMI, and sex on VES-13 scores. DM2, CAD, AF, HF, CKD, OA, fractures, eyes disorders, depression, dementia, PU, and UI were related to higher values of VES-13. Lipid disorders, gastrointestinal diseases, and higher BMI were associated with lower values of VES-13. Blue symbols correspond to odds ratios and 25–75% confidence intervals for the presence of particular disease.

**Figure 2 nutrients-15-03716-f002:**
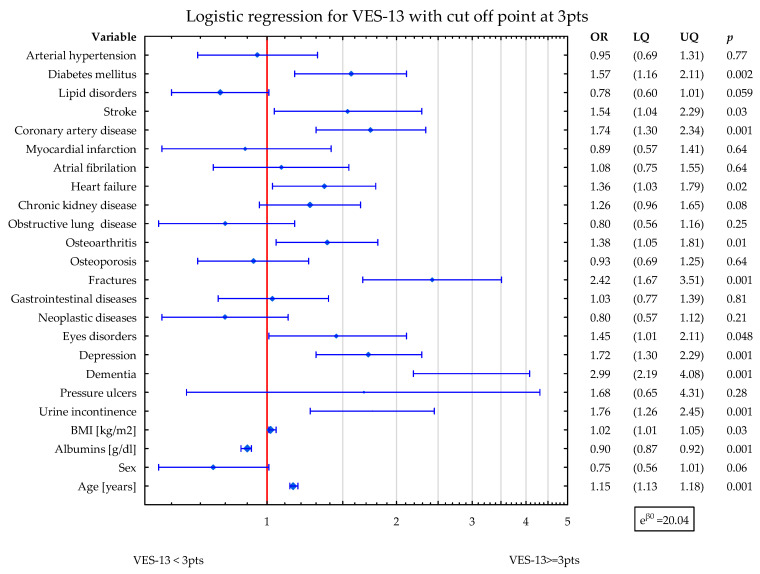
Shows the simultaneous statistical impact of concomitant diseases, BMI, sex, age, and albumins level on VES-13 scores. DM2, stroke, CAD, HF, CKD, OA, fractures, eyes disorders, depression, dementia, and UI were related to higher values of VES-13. The higher albumins level was related to lower values of VES-13. Blue symbols correspond to odds ratios and 25–75% confidence intervals for the presence of particular disease.

**Table 1 nutrients-15-03716-t001:** General characteristics of the study population (n = 2860) according to sex.

Parameter/Disease	Women n = 2036	Men n = 824	*p*-Value
Age (mean ± SD, median (quartiles))	81.7 ± 7.9 83 (77–87)	81.8 ± 8.3 83 (76–88)	*p* = 0.4 ^(U)^
BMI, m/kg^2^ (mean ± SD, median (quartiles))	26.2 ± 5.4 25.4 (22.3–29.2)	26.0 ± 4.6 25.5 (23.0–28.1)	*p* = 0.92 ^(U)^
Body mass, kg (mean ± SD, median (quartiles))	64.4 ± 14.3 63 (54.5–72)	74.8 ± 14.5 73 (65–81)	*p* < 0.0001 ^(U)^
Albumins, g/L (mean ± SD, median (quartiles))	39.7 ± 5.7 41 (36.9–43.7)	39 ± 6.2 40.3 (35.9–43.2)	*p* = 0.002 ^(U)^
VES-13, points (mean ± SD, median (quartiles))	6.78 ± 2.8 8 (4–10)	6.4 ± 2.9 7 (4–9)	*p* = 0.002 ^(U)^
Arterial hypertension, n, %	1618 (79.5%)	633 (76.8%)	*p* = 0.117 ^(chi2)^
Diabetes mellitus, n, %	507 (24.9%)	248 (30.1%)	*p* = 0.004 ^(chi2)^
Lipid disorders, n, %	919 (45.1%)	284 (34.5%)	*p* < 0.0001 ^(chi2)^
Stroke, n, %	322 (15.8%)	141 (17.1%)	*p* = 0.4 ^(chi2)^
Coronary artery disease, n, %	766 (37.6%)	335 (40.7%)	*p* = 0.13 ^(chi2)^
Myocardial infarction, n, %	186 (9.1%)	128 (15.5%)	*p* < 0.0001 ^(chi2)^
Atrial fibrillation, n, %	385 (18.9%)	212 (25.7%)	*p* < 0.0001 ^(chi2)^
Heart failure, n, %	964 (47.4%)	397 (48.2%)	*p* = 0.69 ^(chi2)^
Chronic kidney disease, n, %	1331 (66.1%)	519 (63.6%)	*p* = 0.22 ^(chi2)^
Obstructive lung diseases, n, %	250 (12.3%)	119 (14.4%)	*p* < 0.0001 ^(chi2)^
Osteoarthritis, n, %	767 (37.7%)	212 (25.7%)	*p* < 0.0001 ^(chi2)^
Osteoporosis, n, %	667 (32.8%)	93 (11.3%)	*p* < 0.0001 ^(chi2)^
Fractures, n, %	464 (22.8%)	109 (13.2%)	*p* < 0.0001 ^(chi2)^
Gastrointestinal diseases, n, %	439 (21.5%)	183 (22.2%)	*p* = 0.83 ^(chi2)^
Neoplastic diseases, n, %	293 (14.4%)	153 (18.6%)	*p* = 0.06 ^(chi2)^
Eyes disorders, n, %	302 (14.8%)	101 (12.3%)	*p* = 0.2 ^(chi2)^
Depression, n, %	678 (33.3%)	187 (22.7%)	*p* < 0.0001 ^(chi2)^
Dementia, n, %	866 (42.5%)	313 (38%)	*p* = 0.3 ^(chi2)^
Pressure ulcers, n, %	118 (5.8%)	54 (6.6%)	*p* = 0.96 ^(chi2)^
Urinary incontinence, n, %	443 (21.7%)	114 (13.8%)	*p* < 0.0001 ^(chi2)^

SD: standard deviation, BMI: body mass index, ^(U)^: U Mann-Whitney test, ^(chi2)^: chi-square test.

**Table 2 nutrients-15-03716-t002:** VES-13 according to the presence of the most common diseases.

Disease	Gender	VES-13				*p* Value
		In Patients with the Presence of Particular Disease		In Patients without Particular Disease		
		Mean ± SD	Median (Quartiles)	Mean ± SD	Median (Quartiles)	
Arterial hypertension	Women	6.9 ± 2.8	8.0 (4.0–10.0)	6.2 ± 2.9	7.0 (3.0–8.0)	*p* < 0.0001 ^(U)^
	Men	6.4 ± 2.8	7.0 (3.0–9.0)	6.6 ± 3.0	7.0 (4.0–10.0)	*p* = 0.17 ^(U)^
Diabetes mellitus	Women	7.1 ± 2.7	8.0 (4.0–10.0)	6.7 ± 2.9	7.0 (4.0–10.0)	*p* = 0.01 ^(U)^
	Men	6.5 ± 2.8	7.0 (4.0–9.0)	6.4 ± 2.9	7.0 (3.0–9.0)	*p* = 0.44 ^(U)^
Lipid disorders	Women	6.1 ± 2.9	6.0 (3.0–8.0)	7.3 ± 2.7	8.0 (5.0–10.0)	*p* < 0.0001 ^(U)^
	Men	5.6 ± 2.9	5.5 (3.0–8.0)	6.8 ± 2.8	7.0 (4.0–10.0)	*p* < 0.0001 ^(U)^
Stroke	Women	7.4 ± 2.5	8.0 (6.0–10.0)	6.6 ± 2.9	7.0 (4.0–10.0)	*p* < 0.0001 ^(U)^
	Men	6.9 ± 2.6	7.0 (4.0–9.0)	6.3 ± 2.9	7.0 (3.0–9.0)	*p* = 0.04 ^(U)^
Coronary artery disease	Women	7.4 ± 2.6	8.0 (6.0–10.0)	6.4 ± 2.9	7.0 (3.0–9.0)	*p* < 0.0001 ^(U)^
	Men	6.8 ± 2.8	7.0 (4.0–10.0)	6.1 ± 2.9	7.0 (3.0–8.0)	*p* = 0.0003 ^(U)^
Myocardial infarction	Women	7.3 ± 2.7	8.0 (5.0–10.0)	6.7 ± 2.9	7.0 (4.0–10.0)	*p* = 0.014 ^(U)^
	Men	6.7 ± 2.8	7.0 (4.0–9.5)	6.4 ± 2.9	7.0 (3.0–9.0)	*p* = 0.3 ^(U)^
Atrial fibrillation	Women	7.6 ± 2.6	8.0 (6.0–10.0)	6.6 ± 2.9	7.0 (4.0–9.0)	*p* < 0.0001 ^(U)^
	Men	7.2 ± 2.8	8.0 (5.0–10.0)	6.1 ± 2.8	7.0 (3.0–8.0)	*p* < 0.0001 ^(U)^
Heart failure	Women	7.4 ± 2.6	8.0 (5.0–10.0)	6.2 ± 2.9	7.0 (3.0–9.0)	*p* < 0.0001 ^(U)^
	Men	7.1 ± 2.7	8.0 (5.0–10.0)	5.8 ± 2.9	6.0 (3.0–8.0)	*p* < 0.0001 ^(U)^
Chronic kidney disease	Women	7.2 ± 2.7	7.0 (4.0–10.0)	6.0 ± 3.0	7.0 (3.0–8.0)	*p* < 0.0001 ^(U)^
	Men	6.8 ± 2.8	7.0 (4.0–10.0)	5.7 ± 2.9	6.0 (3.0–8.0)	*p* < 0.0001 ^(U)^
Obstructive lung diseases	Women	7.0 ± 2.7	8.0 (4.0–10.0)	6.7 ± 2.9	8.0 (4.0–10.0)	*p* = 0.24 ^(U)^
	Men	6.6 ± 2.8	7.0 (4.0–9.0)	6.4 ± 2.9	7.0 (4.0–9.0)	*p* = 0.58 ^(U)^
Osteoarthritis	Women	6.8 ± 2.7	8.0 (4.0–10.0)	6.7 ± 2.9	8.0 (4.0–10.0)	*p* = 0.68 ^(U)^
	Men	6.5 ± 2.8	7.0 (4.0–9.0)	6.4 ± 2.9	7.0 (3.5–9.0)	*p* = 0.57 ^(U)^
Osteoporosis	Women	6.4 ± 2.9	7.0 (4.0–9.0)	7.0 ± 2.8	8.0 (4.0–10.0)	*p* = 0.00002 ^(U)^
	Men	6.8 ± 2.8	8.0 (4.0–10.0)	6.4 ± 2.9	7.0 (3.0–9.0)	*p* = 0.15 ^(U)^
Fractures	Women	7.3 ± 2.7	8.0 (4.0–10.0)	6.6 ± 2.9	7.0(4.0–10.0)	*p* < 0.0001 ^(U)^
	Men	7.2 ± 2.6	8.0 (5.0–10.0)	6.3 ± 2.9	7.0(3.0–9.0)	*p* = 0.0017 ^(U)^
Gastrointestinal diseases	Women	6.4 ± 2.8	7.0 (3.0–9.0)	6.9 ± 2.8	8.0 (4.0–10.0)	*p* = 0.0004 ^(U)^
	Men	6.0 ± 2.8	6.0 (3.0–8.0)	6.5 ± 2.9	7.0 (4.0–9.0)	*p* = 0.04 ^(U)^
Neoplastic diseases	Women	6.4 ± 2.9	7.0 (4.0–9.0)	6.8 ± 2.8	8.0 (4.0–10.0)	*p* = 0.04 ^(U)^
	Men	6.8 ± 2.8	8.0 (4.0–9.0)	6.3 ± 2.9	7.0 (3.0–9.0)	*p* = 0.07 ^(U)^
Eyes disorders	Women	7.0 ± 2.7	8.0 (4.0–10.0)	6.7 ± 2.9	8.0 (4.0–10.0)	*p* = 0.14 ^(U)^
	Men	7.0 ± 2.6	7.0 (5.0–9.0)	6.3 ± 2.9	7.0 (3.0–9.0)	*p* = 0.37 ^(U)^
Depression	Women	7.2 ± 2.6	8.0 (6.0–10.0)	6.5 ± 2.9	7.0 (4.0–10.0)	*p* < 0.0001 ^(U)^
	Men	7.0 ± 2.7	8.0 (4.0–9.0)	6.2 ± 2.9	7.0 (3.0–9.0)	*p* = 0.002 ^(U)^
Dementia	Women	8.0 ± 2.3	8.0 (7.0–10.0)	5.9 ± 2.9	6.0 (3.0–8.0)	*p* < 0.0001 ^(U)^
	Men	7.7 ± 2.5	8.0 (7.0–10.0)	5.6 ± 2.8	5.0 (3.0–8.0)	*p* < 0.0001 ^(U)^
Pressure ulcers	Women	8.5 ± 2.0	9.0 (8.0–10.0)	6.7 ± 2.8	7.0 (4.0–10.0)	*p* < 0.0001 ^(U)^
	Men	8.4 ± 1.8	8.5 (7.0–10.0)	6.3 ± 2.9	7.0 (3.0–9.0)	*p* < 0.0001 ^(U)^
Urinary incontinence	Women	6.9 ± 2.6	8.0 (4.0–9.0)	7.0 ± 2.9	8.0 (4.0–10.0)	*p* = 0.53 ^(U)^
	Men	7.0 ± 2.4	7.0 (5.0–9.0)	6.3 ± 3.0	7.0 (3.0–9.0)	*p* = 0.033 ^(U)^

SD: standard deviation, ^(U)^: U Mann-Whitney test.

**Table 3 nutrients-15-03716-t003:** Age of patients according to the presence of the most common diseases.

Disease	Gender	Age				*p* Value
		In Patients with the Presence of Particular Disease		In Patients without Particular Disease		
		Mean ± SD	Median (Quartiles)	Mean ± SD	Median (Quartiles)	
Arterial hypertension	Women	82.3 ± 7.5	83 (78–87)	79.3 ± 8.8	80 (72–86)	*p* < 0.0001 ^(U)^
	Men	81.8 ± 8.2	83 (76–88)	81.8 ± 8.5	83 (76–88)	*p* = 0.81 ^(U)^
Diabetes mellitus	Women	81.9 ± 7.4	83 (77–87)	81.6 ± 8.1	83 (77–87)	*p* = 0.9 ^(U)^
	Men	80.8 ± 8.2	82 (74–87)	82.3 ± 8.3	84 (78–88)	*p* = 0.03 ^(U)^
Lipid disorders	Women	79.9 ± 7.9	81 (74–86)	83.2 ± 7.6	84 (79–88)	*p* < 0.0001 ^(U)^
	Men	80.1 ± 8.6	82 (74–86)	82.8 ± 8.0	84 (79–88)	*p* < 0.0001 ^(U)^
Stroke	Women	82.7 ± 7.5	84 (79–88)	81.5 ± 7.9	83 (77–87)	P = 0.02 ^(U)^
	Men	81.1 ± 8.3	83 (75–88)	82.0 ± 8.3	83 (77–88)	*p* = 0.3 ^(U)^
Coronary artery disease	Women	83.4 ± 6.8	84 (79–88)	80.7 ± 8.3	82 (75–87)	*p* < 0.0001 ^(U)^
	Men	83.7 ± 7.7	85 (80–89)	80.6 ± 8.4	82 (75–87)	*p* < 0.0001 ^(U)^
Myocardial infarction	Women	83.1 ± 7.3	85 (79–88)	81.6 ± 7.9	83 (77–87)	*p* = 0.007 ^(U)^
	Men	83.5 ± 7.3	85 (79.5–88)	81.5 ± 8.4	83 (76–88)	*p* = 0.02 ^(U)^
Atrial fibrillation	Women	84.2 ± 6.6	85 (80–89)	81.1 ± 8.1	82 (76–87)	*p* < 0.0001 ^(U)^
	Men	84.2 ± 7.6	86 (81–89)	81.0 ± 8.4	82 (75–87)	*p* < 0.0001 ^(U)^
Heart failure	Women	83.8 ± 6.9	85 (80–88)	79.8 ± 8.3	81 (74–86)	*p* < 0.0001 ^(U)^
	Men	83.6 ± 7.8	85 (80–89)	80.1 ± 8.3	81 (74–86)	*p* < 0.0001 ^(U)^
Chronic kidney disease	Women	83.8 ± 6.7	84 (80–88)	77.6 ± 8.4	78 (71–84)	*p* < 0.0001 ^(U)^
	Men	84.5 ± 6.6	85 (81–89)	77.3 ± 9.0	78 (70–84)	*p* < 0.0001 ^(U)^
Obstructive lung diseases	Women	81.9 ± 7.0	82.5 (78–87)	81.7 ± 8.0	83 (77–87)	*p* = 0.9 ^(U)^
	Men	82.2 ± 7.5	83 (78–87)	81.8 ± 8.4	83 (76–88)	*p* = 0.9 ^(U)^
Osteoarthritis	Women	81.9 ± 7.4	83 (78–87)	81.6 ± 8.2	83 (76–88)	*p* = 0.68 ^(U)^
	Men	82.0 ± 8.4	84 (78–88)	81.8 ± 8.2	83 (76–88)	*p* = 0.9 ^(U)^
Osteoporosis	Women	80.9 ± 7.9	82 (76–87)	82.1 ± 7.9	83 (77–88)	*p* = 0.001 ^(U)^
	Men	83.5 ± 7.3	84 (79–89)	81.6 ± 8.4	83 (76–88)	*p* = 0.06 ^(U)^
Fractures	Women	82.6 ± 7.3	84 (78–88)	81.4 ± 8.0	83 (76.5–87)	*p* = 0.009 ^(U)^
	Men	83.3 ± 9.6	84 (78–91)	81.6 ± 8.1	83 (76–88)	*p* = 0.02 ^(U)^
Gastrointestinal diseases	Women	79.8 ± 8.4	81 (74–86)	82.2 ± 7.7	83 (78–88)	*p* < 0.0001 ^(U)^
	Men	81.5 ± 7.6	83 (77–87)	81.9 ± 8.5	83 (76–88)	*p* = 0.5 ^(U)^
Neoplastic diseases	Women	81.0 ± 8.0	82 (76–87)	81.8 ± 7.9	83 (77–87)	*p* = 0.07 ^(U)^
	Men	83.0 ± 7.9	84 (80–88)	81.6 ± 8.4	83 (76–88)	*p* = 0.06 ^(U)^
Eyes disorders	Women	82.6 ± 7.3	83 (79–88)	81.6 ± 8.0	83 (77–87)	*p* = 0.2 ^(U)^
	Men	83.7 ± 7.4	85 (81–89)	81.6 ± 8.4	83 (76–88)	*p* = 0.01 ^(U)^
Depression	Women	81.8 ± 7.3	83 (78–87)	81.7 ± 8.2	83 (77–88)	*p* = 0.7 ^(U)^
	Men	81.9 ± 7.6	83 (78–87)	81.8 ± 8.5	83 (76–88)	*p* = 0.8 ^(U)^
Dementia	Women	84.6 ± 6.4	85 (81–89)	79.5 ± 8.2	81 (73–86)	*p* < 0.0001 ^(U)^
	Men	84.1 ± 7.2	85 (80–89)	80.4 ± 8.6	82 (74–87)	*p* < 0.0001 ^(U)^
Pressure ulcers	Women	84.4 ± 7.9	85 (80–90)	81.5 ± 7.9	83 (77–87)	*p* = 0.0002 ^(U)^
	Men	83.7 ± 6.9	85 (82–88)	81.7 ± 8.4	83 (76–88)	*p* = 0.07 ^(U)^
Urinary incontinence	Women	81.8 ± 7.6	83 (77–87)	81.6 ± 8.0	83 (77–87)	*p* = 0.9 ^(U)^
	Men	82.5 ± 7.8	84 (78–88)	81.7 ± 8.3	83 (76–88)	*p* = 0.3 ^(U)^

SD: standard deviation, ^(U)^: U Mann-Whitney test.

## Data Availability

The statistical data used to support the presented findings may be obtained upon request to corresponding author.

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
