# Peer review of "Cardiometabolic Disorders Are Important Correlates of Vulnerability in Hospitalized Older Adults"

_nutrients, 2023, doi:10.3390/nu15173716_

Round 1

Reviewer 1 Report

Cardiometabolic disorders as important correlates of vulnera- 2 bility in hospitalized older adults

Manuscript entitled “Cardiometabolic disorders as important correlates of vulnera- 2 bility in hospitalized older adults” by Ganna et al., is a good study. Here, the authors assess the association between the Vulnerable Elders Survey-13 (VES-13) score among the 2860 participants with an average age of 83 years. Multiple logistic regression with adjustment for age and nutritional status was used to assess the independent impact of every disease on vulnerability. Diabetes mellitus type 2, coronary artery disease, atrial fibrillation, heart failure, chronic kidney disease, osteoarthritis, fractures, eyes disorders, depression, dementia, pressure ulcers, and urinary incontinence were associated with higher scores of VES-13.

Overall, the information presented in this research article is useful, but I approve of its publication after modifications.

1.       Manuscript need to be rechecked for grammatical mistakes such as at line -88, the total total number of individuals was selected – it should be were

2.       For Figure 1 & Figure 2 the Image is not clear.

3.       Did you use any other correlations such as Spearman correlation? Is it showing the same stuff that what VES-13 is showing?

Cardiometabolic disorders as important correlates of vulnera- 2 bility in hospitalized older adults

Manuscript entitled “Cardiometabolic disorders as important correlates of vulnera- 2 bility in hospitalized older adults” by Ganna et al., is a good study. Here, the authors assess the association between the Vulnerable Elders Survey-13 (VES-13) score among the 2860 participants with an average age of 83 years. Multiple logistic regression with adjustment for age and nutritional status was used to assess the independent impact of every disease on vulnerability. Diabetes mellitus type 2, coronary artery disease, atrial fibrillation, heart failure, chronic kidney disease, osteoarthritis, fractures, eyes disorders, depression, dementia, pressure ulcers, and urinary incontinence were associated with higher scores of VES-13.

Overall, the information presented in this research article is useful, but I approve of its publication after modifications.

1.       Manuscript need to be rechecked for grammatical mistakes such as at line -88, the total total number of individuals was selected – it should be were

2.       For Figure 1 & Figure 2 the Image is not clear.

3.       Did you use any other correlations such as Spearman correlation? Is it showing the same stuff that what VES-13 is showing?

Author Response

   Review 1

We do thank the Reviewer for all the constructive comments.

  1. Manuscript need to be rechecked for grammatical mistakes such as at line -88, the total total number of individuals was selected – it should be were

The manuscript has been edited by a native English person.

  1. For Figure 1 & Figure 2 the Image is not clear.

Figures have been re-drafted for better presentation

  1. Did you use any other correlations such as Spearman correlation? Is it showing the same stuff that what VES-13 is showing?

Response: Pearson correlation coefficients were used. With such large samples, Spearman correlations gave virtually the same results.

Linear correlation was performed for age and VES-13 with R=0.587 (p<0.001), BMI and VES-13 with R=-0.11 (p<0.001), and albumins and VES-13 with inverse correlation, R=-0.37 (p<0.001).

Reviewer 2 Report

The authors report a retrospective analysis to evaluate factors associated with "vulnerability" in hospitalized older patients at a single center in Poland. Unsurprisingly, the authors found that several conditions were associated with a higher level of vulnerability in these patients.

The topic of the article is not particularly new, and may wonder whether this fit the scope of the journal. The main issue of this study, however, is the merely speculative/descriptive nature of their findings, which clinical relevance is unclear.

In details:

- In the introduction the authors state that "there is a gap of knowledge in the cardiometabolic diseases' impact on VES-13 and frailty". I feel that the association between "cardiometabolic" conditions and frailty has been previously, extensively and repeatedly described: these are well known contributors and drivers of frailty, and one may find perhaps unsurprising that such conditions are associated with frailty in older patients. In short, the novel bit of this investigation seem unclear and should better exposed.

- Paragraph 2.1 - detailed reasons for exclusion of patients should be reported. I suppose that most patients excluded were because of lack of VES-13 data; one may wonder whether there were some factors that led to lack of VES-13 questionnaire in these patients, which could have alter the results observed. In other words, one may suspect that data regarding VES-13 do not miss at random, but rather miss due to reasons associated with some specific conditions (for example, severity of the underlying disease? dementia?). If so, this could have really influenced the results observed. A table reporting patients excluded and included, and their baseline characteristics, could be useful to have a hint on that.

- The retrospective assessment of the cardiometabolic condition is a clear limitation of this study. Indeed, I suppose that if the conditions was not clearly reported in the medical chart, this was flagged as "absent" - nonetheless, some of these conditions could have been incompletely reported in the medical charts of these patients (I am looking specifically at osteoarthritis, COPD, previous factors, depression etc.). This could have clearly influenced the findings observed and reported in this study.

- Authors reported to have used the "BIS1" formula to calculate eGFR in this cohort - this is definitely not one of the most used equation to calculate eGFR, especially when compared to cockroft-gault, MDRD or CKD-EPI. Can the authors justify why they used such formula?

- Main concern of this study is related to the methodology of the analysis. There are different issues which may need clarification or amendment. First, it is unclear why the authors performed correlation analysis if their aim was to analyse if the cardiometabolic conditions were associated with VES-13 (i.e. frailty). Indeed, a regression is probably more appropriate in doing so, and the added value of the correlation analysis seem unclear.

- Another issue is related to the dichotomization of VES-13. It is unclear whether the cut-off chosen was previously validated to define frailty, and how the authors selected this; of note, dichotomising continuous variable is currently not considered as best practice, as this imply a strong difference between the "above" and "below" values of such variables; which is most commonly never the case in medicine (i.e., in this example, this assume that patients with VES-13 of 2 vs. 3 were more different that those with VES-13 of 3 vs. 8; and that is clearly not true).

- There are other issues with the logistic regression model: first, there is no information on how the candidate variables were selected; second, it is unclear if the results reported in the Figures are related to a single model including all the variables at the same time, or instead if the OR and 95%CI are instead related to different models (one for each variable of interest), each of the model adjusted for age, sex etc. Can the authors clarify? Also, please note that when reporting logistic regression model, Intercept should be reported along with the other covariates.

- How were continuous variable handled in such logistic regression model? In other words, were these dicohotomised or not? Was non-linear association of such variables with the dependent variable assessed and, if present, handled appropriately? We miss this data in this version of the manuscript.

- In the results, the authors seem to suggest that the results implies "increased odds ratio for risk of death and disability" (see for example lines 185-186 in the PDF). This comes particularly unexpected, as the authors are not analysing whatsoever the risk of death or disability, but instead performing a cross-sectional analysis on the association between some variables and the VES-13 > 3; one could definitely not conclude that such OR are for increased odds of death and disability, as this is merely speculative. Can the authors clarify such statements? I feel as these are not appropriate and not supported by the data shown by the authors.

- I can't find Table 2 and Table 3 particularly informative: these tables could be placed in supplementary materials if needed, but the information provided is not very informative, especially when considering the median values (which are all similar given that this is overly influenced by the underlying distribution of the VES-13 in the overall population.

Please check carefully the english language of this article as there seem to be some mistakes (see for example last line of the abstract, "what may be attributed to younger age" - perhaps "what" is not correct in this sentence? There are other example throughout the manuscript).

Author Response

Review 2

We do thank the Reviewer for all the constructive comments.

The authors report a retrospective analysis to evaluate factors associated with "vulnerability" in hospitalized older patients at a single center in Poland. Unsurprisingly, the authors found that several conditions were associated with a higher level of vulnerability in these patients.

The topic of the article is not particularly new, and may wonder whether this fit the scope of the journal. The main issue of this study, however, is the merely speculative/descriptive nature of their findings, which clinical relevance is unclear.

Response: We do agree that there are studies showing the relationship between single conditions and higher level of vulnerability in older adults. Nevertheless, literature lacks studies on complex association and contribution of different diseases to overall vulnerability in a “real word” large population of geriatric patients.

Therefore, we believe there is substantial clinical relevance in the conducted study. It provides insights into which diseases are more strongly associated with vulnerability in hospitalized patients of advanced age. For example, our work demonstrates that conditions such as a history of myocardial infarction are not linked to higher VES-13 scores. Meanwhile, eye disorders or urinary incontinence are associated with higher (worse) test outcomes. In clinical practice, this could aid in directing attention toward diseases correlated with elevated VES-13 scores.

In details:

- In the introduction the authors state that "there is a gap of knowledge in the cardiometabolic diseases' impact on VES-13 and frailty". I feel that the association between "cardiometabolic" conditions and frailty has been previously, extensively and repeatedly described: these are well known contributors and drivers of frailty, and one may find perhaps unsurprising that such conditions are associated with frailty in older patients. In short, the novel bit of this investigation seem unclear and should better exposed.

Response: Our primary focus was on publications concerning the VES-13 assessment tool and various diseases, predominantly those of cardiometabolic nature. Data on the impact of different diseases on VES-13 scores were extensively represented in relation to oncological conditions and qualification for surgical interventions. Two publications evaluating the prevalence of vulnerability in patients with acute coronary syndrome and/or decompensated heart failure were found. In cases where publications using the VES-13 were absent, data regarding the association of different diseases with frailty assessment scales such as the Frailty Index or Fried's phenotypic definition of frailty were often found. However, assessing the influence of diseases on the syndrome evaluated through different criteria proves challenging. Furthermore, VES-13 stands as an easy and quick scale recommended for routine practical application (qualification for comprehensive geriatric assessment, chemotherapy, surgical interventions). Hence, we intended  to show concomitantly the impact of various diseases on vulnerability specifically within the framework of this scale.

According to the suggestion of the reviewer, the novelty and clinical relevance of this approach have been highlighted in the introduction and further developed in the discussion.

- Paragraph 2.1 - detailed reasons for exclusion of patients should be reported. I suppose that most patients excluded were because of lack of VES-13 data; one may wonder whether there were some factors that led to lack of VES-13 questionnaire in these patients, which could have alter the results observed. In other words, one may suspect that data regarding VES-13 do not miss at random, but rather miss due to reasons associated with some specific conditions (for example, severity of the underlying disease? dementia?). If so, this could have really influenced the results observed. A table reporting patients excluded and included, and their baseline characteristics, could be useful to have a hint on that.

Response: The situation where older hospitalized patients are being excluded from given analyses due to reasons associated with some specific conditions like severity of the underlying disease or dementia is typical for geriatric studies.

In the present study, all 3382 older patients that had been admitted to the Department of Geriatrics during 2012-2019 were screened. The criterion for participation in this study was efficient verbal communication. Out of the 3382 patients, 128 were readmitted to the Department of Geriatrics and were excluded from further analyses. Further 394 patients were excluded because of severe dementia or terminal illness. After screening, 2860 patients (2036 women and 824 men) met the inclusion criteria and were enrolled in the analysis.

This has been presented more clearly in the methods and limitations sections of the study.

- The retrospective assessment of the cardiometabolic condition is a clear limitation of this study. Indeed, I suppose that if the conditions was not clearly reported in the medical chart, this was flagged as "absent" - nonetheless, some of these conditions could have been incompletely reported in the medical charts of these patients (I am looking specifically at osteoarthritis, COPD, previous factors, depression etc.). This could have clearly influenced the findings observed and reported in this study.

Response: Upon admission, detailed medical anamnesis and examination of medical history are  performed. Furthermore, Comprehensive Geriatric Assessment is performed with each patient. Therefore, it is extremely unlikely that any information may be missing.

This information has been provided more explicitly. The above has been provided more explicitly in the paper.

- Authors reported to have used the "BIS1" formula to calculate eGFR in this cohort - this is definitely not one of the most used equation to calculate eGFR, especially when compared to cockroft-gault, MDRD or CKD-EPI. Can the authors justify why they used such formula?

Response: The Berlin Initiative Study (BIS) has been developed to be used in older people, and has been validated against gold-standard measured GFR:

Schaeffner, E.S.; Ebert, N.; Delanaye, P.; Frei, U.; Gaedeke, J.; Jakob, O.; Kuhlmann, M.K.; Schuchardt, M.; Tolle, M.; Ziebig, R.; et al. Two novel equations to estimate kidney function in persons aged 70 years or older. Ann. Intern. Med. 2012, 157, 471–481.

Alshaer, I.M.; Kilbride, H.S.; Stevens, P.E.; Eaglestone, G.; Knight, S.; Carter, J.L.; Delaney, M.P.; Farmer, C.K.; Irving, J.; O’Riordan, S.E.; et al. External validation of the Berlin equations for estimation of GFR in the elderly. Am. J. Kidney Dis. 2014, 63, 862–865

Da Silva Selistre, L.; Rech, D.L.; de Souza, V.; Iwaz, J.; Lemoine, S.; Dubourg, L. Diagnostic Performance of Creatinine-Based Equations for Estimating Glomerular Filtration Rate in Adults 65 Years and Older. JAMA Intern. Med. 2019, 179, 796–804.

Therefore, this formula is routinely used by our diagnostic laboratory.

- Main concern of this study is related to the methodology of the analysis. There are different issues which may need clarification or amendment. First, it is unclear why the authors performed correlation analysis if their aim was to analyse if the cardiometabolic conditions were associated with VES-13 (i.e. frailty). Indeed, a regression is probably more appropriate in doing so, and the added value of the correlation analysis seem unclear.

Response: In bivariate analyses, qualitative variables were compared using the Chi-square test, quantitative variables - with the Mann-Whitney U-test. Correlation analysis was performed solely for age and VES-13, BMI and VES-13, as well as albumin levels and VES-13, serving as preliminary assessments prior to constructing a second model incorporating these variables. Then, we employed multivariate logistic regression.

- Another issue is related to the dichotomization of VES-13. It is unclear whether the cut-off chosen was previously validated to define frailty, and how the authors selected this; of note, dichotomising continuous variable is currently not considered as best practice, as this imply a strong difference between the "above" and "below" values of such variables; which is most commonly never the case in medicine (i.e., in this example, this assume that patients with VES-13 of 2 vs. 3 were more different that those with VES-13 of 3 vs. 8; and that is clearly not true).

Response: Based on the findings of the VES-13 scale authors (Saliba D. et al., 2001), patients with scores of ≥3 points exhibited a 4.2-fold increased risk of mortality or functional deterioration over 2-year period compared  to individuals with scores <3. In light of this, the authors recommend utilizing a score of ≥3 as the designated threshold or cutoff point.

Predictive value of VES-13 was further confirmed in many studies, e.g. the VES-13 strongly predicted death and functional decline or medical services use:

Mohile SG, Bylow K, Dale W, Dignam J, Martin K, Petrylak DP, Stadler WM, Rodin MB: A pilot study of the vulnerable elders survey-13 compared with the comprehensive geriatric assessment for identifying disability in older patients with prostate cancer who receive androgen ablation. Cancer 2007;109(4):802-810

Min LC, Elliott MN, Wenger NS, Saliba D: Higher vulnerable elders survey scores predict death and functional decline in vulnerable older people. J Am Geriatr Soc 2006;54(3):507-511

Min L, Yoon W, Mariano J, Wenger NS, Elliott MN, Kamberg C, Saliba D: The vulnerable elders-13 survey predicts 5-year functional decline and mortality outcomes in older ambulatory care patients. J Am Geriatr Soc 2009;57(11):2070-2076

McGee HM, O'Hanlon A, Barker M, Hickey A, Montgomery A, Conroy R, O'Neill D: Vulnerable older people in the community: relationship between the Vulnerable Elders Survey and health service use. J Am Geriatr Soc 2008;56(1):8-15

We employed dichotomization of the VES-13 results for constructing the logistic regression. However, during the bivariate analysis, we treated VES-13 as a continuous variable to circumvent the threshold-related influence you mentioned.

- There are other issues with the logistic regression model: first, there is no information on how the candidate variables were selected; second, it is unclear if the results reported in the Figures are related to a single model including all the variables at the same time, or instead if the OR and 95%CI are instead related to different models (one for each variable of interest), each of the model adjusted for age, sex etc. Can the authors clarify? Also, please note that when reporting logistic regression model, Intercept should be reported along with the other covariates.

Response: Diseases, which expressed significance in bivariable analysis, were employed in the logistic regression model including concurrently all the diseases significantly related to VES-13 with Mann-Whitney U-test. To assess the potential influence of diseases on VES-13 simultaneously, two statistical models were constructed. The first one considered sex and BMI, while the second one also included age and albumins level.

This information has been completed in the Statistical analysis section of the study and more clearly presented in the Results. Intercept has also been reported, as suggested by the Reviewer.

- How were continuous variable handled in such logistic regression model? In other words, were these dicohotomised or not? Was non-linear association of such variables with the dependent variable assessed and, if present, handled appropriately? We miss this data in this version of the manuscript.

Response: Continuous variables (age, BMI and albumins) were entered to the logistic regression model as continuous variables, not dichotomized. VES-13 was dichotomized, as explained above. This has been stated in the Statistical analysis section of the study and more clearly presented in the Results.

Bivariate relationship of continuous variables (age, BMI and albumins) with VES-13 has been presented in the Results.

- In the results, the authors seem to suggest that the results implies "increased odds ratio for risk of death and disability" (see for example lines 185-186 in the PDF). This comes particularly unexpected, as the authors are not analysing whatsoever the risk of death or disability, but instead performing a cross-sectional analysis on the association between some variables and the VES-13 > 3; one could definitely not conclude that such OR are for increased odds of death and disability, as this is merely speculative. Can the authors clarify such statements? I feel as these are not appropriate and not supported by the data shown by the authors.

Response: As a matter of fact, "increased odds ratio for risk of death and disability" comes from previous studies with VES-13 and wasn’t analyzed in the present report. Relevant changes were performed and this phrase has been deleted.

- I can't find Table 2 and Table 3 particularly informative: these tables could be placed in supplementary materials if needed, but the information provided is not very informative, especially when considering the median values (which are all similar given that this is overly influenced by the underlying distribution of the VES-13 in the overall population.

We do agree that median values look rather similar. Medians have been given to show lege artis values used for Mann-Whitney U-test. We also provided mean ± standard deviation values for better presenting  potential differences.

Both tables (especially Table 2) show, in our opinion,  essential information on the relationship of VES-13 and age to the prevalence of the most common diseases in hospitalized older adults.

Nevertheless, we may place (especially Table 3) in the supplementary materials.

Round 2

Reviewer 2 Report

Authors have sufficiently answered all my previous comments, and I think the manuscript has improved compared to previous version.

Issues remains in terms of novelty and clinical significance; the authors have expanded in their manuscript on why their results appear useful and of clinical relevance.

I have no further comments.